# A unified theory for the origin of grid cells through the lens of pattern formation

**Ben Sorscher\*[1], Gabriel C. Mel\*[2], Surya Ganguli[1], Samuel A. Ocko[1]**
[1]Department of Applied Physics, Stanford University
[2]Neurosciences PhD Program, Stanford University

## Abstract

Grid cells in the brain fire in strikingly regular hexagonal patterns across space. There are currently two seemingly unrelated frameworks for understanding these patterns. *Mechanistic models* account for hexagonal firing fields as the result of pattern-forming dynamics in a recurrent neural network with hand-tuned center-surround connectivity. *Normative models* specify a neural architecture, a learning rule, and a navigational task, and observe that grid-like firing fields emerge due to the constraints of solving this task. Here we provide an analytic theory that unifies the two perspectives by casting the learning dynamics of neural networks trained on navigational tasks as a pattern forming dynamical system. This theory provides insight into the optimal solutions of diverse formulations of the normative task, and shows that symmetries in the representation of space correctly predict the structure of learned firing fields in trained neural networks. Further, our theory proves that a nonnegativity constraint on firing rates induces a *symmetry-breaking* mechanism which favors hexagonal firing fields. We extend this theory to the case of learning multiple grid maps and demonstrate that optimal solutions consist of a hierarchy of maps with increasing length scales. These results unify previous accounts of grid cell firing and provide a novel framework for predicting the learned representations of recurrent neural networks.

## 1 Introduction

How does the brain construct an internal map of space? One such map is generated by grid cells in the medial entorhinal cortex (MEC), which exhibit regular hexagonal spatial firing fields, forming a periodic, low-dimensional representation of space [1]. Grid cells are clustered into discrete modules sharing a periodicity and an orientation, but varying randomly in phase [1, 2]. A complementary map is generated by place cells in the adjacent hippocampus, which exhibit localized spatial firing fields, forming a sparser representation of space [3].

Early mechanistic models of grid cells corresponded to recurrent neural networks (RNNs) with hand-tuned connectivity designed specifically to reproduce hexagonal grid cell firing patterns [4, 5, 6]. Such continuous attractor models can robustly integrate and store 2D positional information via path integration [7]. Recent enhancements to such attractor networks that incorporate plastic inputs from landmark cells can explain why grid cells deform in irregular environments [8], and when they either phase shift or remap in altered virtual reality environments [9]. However, none of these recurrent network models show that grid-like firing patterns are *required* to solve navigational tasks. Thus they cannot demonstrate that hexagonal firing patterns naturally arise as the optimal solution to any computational problem, precisely because the hexagonal patterns are simply assumed in the first place by hand-tuning the recurrent connectivity.

More recent normative models have shown that neural networks trained on tasks that involve encoding a representation of spatial position learn grid-like responses in their hidden units. For example,

[10] found that the weights of a one layer neural network, trained via Oja's rule on simulated place cell inputs, learned grid cells with square grid firing patterns. When a non-negativity constraint was imposed, these grids became hexagonal. In [11], the eigenvectors of the graph Laplacian for a navigational task were shown to be square-like grids. [12] showed that learning basis functions for local transition operators also yields square grids. [13, 14] trained recurrent neural networks on a path integration task, and observed that the hidden units developed grid-like patterns. [13] found square grids in square environments and hexagonal grids in triangular environments. [14] claimed to find hexagonal grids, though their resulting patterns had substantial heterogeneity. While these normative models hint at the intriguing hypothesis that grid like representations in the brain may arise as an inevitable consequence of solving a spatial navigation task , these models alone do not offer any theoretical clarity on *when* and *why* grid cell patterns emerge from such navigational solutions, and if they do, why they are sometimes square, sometimes hexagonal, or sometimes highly heterogeneous [1]

Here we provide such theoretical clarity by forging an analytic link between the learning dynamics of neural networks trained on navigational tasks, to a central, unifying pattern forming dynamical system. Our theory correctly predicts the structure and hierarchy of grid patterns learned in diverse neural architectures, and proves that nonnegativity is just one of a family of interpretable representational constraints that promotes hexagonal grids. Furthermore, this theory unifies *both* mechanistic and normative models, by proving that the learning dynamics induced by optimizing a normative position encoding objective is equivalent to the mechanistic pattern forming dynamics implemented by a population of recurrently connected neurons.

## 2  Optimally encoding position yields diverse grid-like patterns

To study the space of solutions achieved by the normative models outlined above, we train a variety of neural network architectures on navigational tasks, reproducing the work of [10, 13, 14] (Fig. 1A). We simulate an animal moving in a square box, 2.2m on a side, and record the activity of $n_P$ simulated place cells tiled randomly and uniformly throughout the environment (Fig. 1B). We collect the place cell activations at $n_x$ locations as the animal explores its environment in a matrix $P \in \mathbb{R}^{n_x \times n_P}$. We then train the following network architectures on the following tasks:

1. **1-layer NN**. Following [10], we train a single layer neural network to perform unsupervised Hebbian learning on place cell activations $P$ as inputs. Hidden unit representations are made orthogonal by a generalized Hebbian algorithm similar to Gram-Schmidt orthogonalization (see [10] for details). This learning procedure is equivalent to performing PCA on place cell inputs.

2. **RNN**. We train an RNN to encode position by path integrating velocity inputs. At each time step, the network receives the animal's 2-dimensional velocity $\vec{v}(t)$ as input. The velocity signal is integrated by the network's $n_G$ recurrently connected units, and the network's current position representation is linearly read out into a layer of estimated place cells. This approach is identical to that used in [13], except that our RNNs are trained to encode position by encoding in its outputs a place cell representation rather than a 2D vector of Cartesian $(x, y)$ position.

3. **LSTM**. We train a significantly more complex LSTM architecture on the same path integration task as in 2, reproducing the work of [14]. The "grid cells" in this architecture are not recurrently connected, but reside in a dense layer immediately following the LSTM. The grid cells are also subject to dropout at a rate of $0.5$. We train both with and without the additional objective of integrating head direction inputs, and obtain qualitatively similar results.

Remarkably, in each case the networks learn qualitatively similar grid-like representations (Fig. 1C-E). We observe that the structure of the grid patterns depends sensitively on the shape of the place cell tuning curves. We first train with Gaussian place cell tuning curves of size 400cm$^2$, and find that each network develops regular square grid patterns (Fig. 1C), like those in [13]. We next train with a center-surround place cell tuning curve, like that used in [10], and find that each network

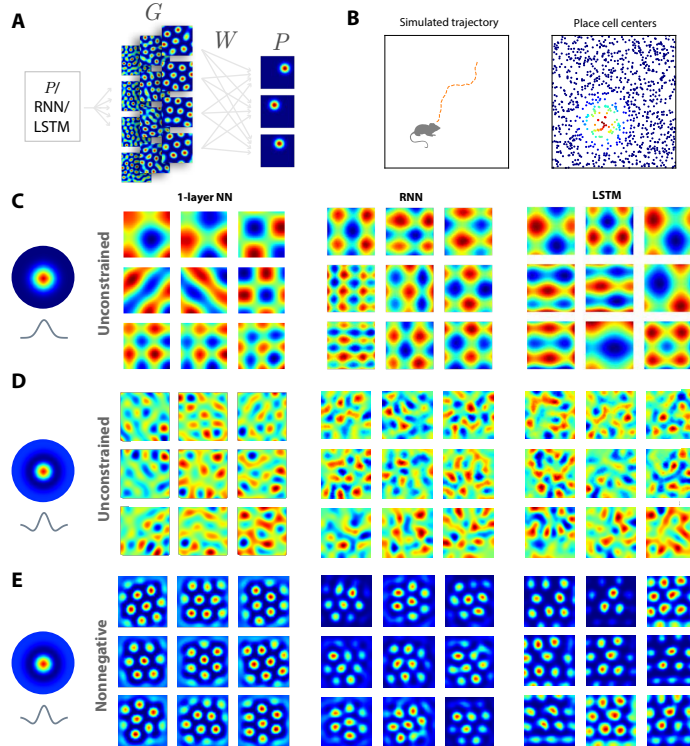

Figure 1: Neural networks trained on normative tasks develop grid-like firing fields. (A) A schematic of the position encoding objective. Depending on the task, $G$ may receive external inputs directly from place cells (as in [10]), or from an RNN or an LSTM that in turn only receives velocity inputs (as in [13, 14]). In the latter case, the recurrent net must generated hidden representations $G$ that convert velocity inputs to a place code $P$ through a set of trained read-out weights $W$. (B) Left: simulated animal trajectory. Right: the place cell centers (dots) of the desired place code $P$ in (A) uniformly and isotropically tile the environment. Blue to red indicate low to high firing rates when the animal is at the location on the left. (C-E) From left to right, we train a single layer neural network, an RNN, and an LSTM on place cell outputs, reproducing the results of [10, 13, 14]. C) When the place cell receptive field (left) is broad, all networks learn square grids. D) When the place cell receptive field is a difference of gaussians, all networks learn amorphous, quasi-periodic patterns. E) When a nonnegativity constraint is imposed on hidden unit activations, all networks now learn regular hexagonal grids.

develops amorphous, quasi-periodic patterns (Fig. 1D). These patterns have grid-like features, and occasionally appear hexagonal, much like the patterns found in [14]. We also note that we obtained similar amorphous patterns when we trained with the tuning curves used in [14], which are extremely sharply peaked ($1cm^2$), so that the place cell code is approximately one-hot (Fig. 6).

**A non-negativity constraint induces hexagonal grids.** [10] observed that imposing a non-negativity constraint on the output of their 1-layer neural network changed the learned grid patterns from square to hexagonal. However, since such feedforward networks only convert place cell *inputs* to outputs learned in an unsupervised manner, it is *a priori* unclear how non-negativity might impact the internal representations of recurrent neural networks trained to convert *velocity inputs* to place cell *outputs*. To investigate whether the same constraint would alter the structure of the patterns observed in the other architectures and in more complex navigational tasks, we retrain all architectures while imposing non-negativity on the activations of hidden units that ultimately give rise to grid cells. We find that this constraint consistently yields regular hexagonal grids in each architecture (Fig. 1E).

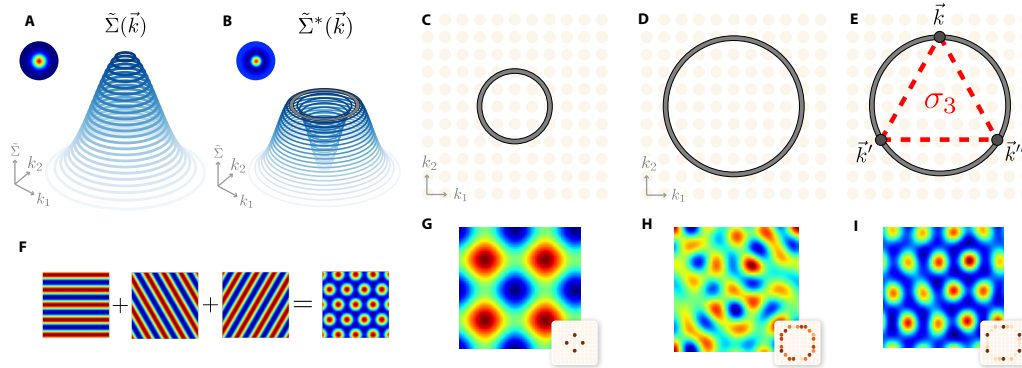

Figure 2: Pattern formation theory predicts structure of learned representations. A,B) Visualization of $\tilde{\Sigma}$ and $\tilde{\Sigma}^*$. D-E) The top subspace of $\tilde{\Sigma}^*$ is a degenerate ring. Absent any other constraint, pattern forming dynamics will yield arbitrary combinations of Fourier modes on this ring. B) When the ring is close to zero, only $90°$ combinations of modes are available due to discretization effects, yielding square lattices. C) When the ring is larger, many degenerate modes are available. E) The softened nonnegativity constraint of Eq. 6 induces a three-body interaction between triplets of spatial frequencies which add to zero. Within the top subspace of $\tilde{\Sigma}^*$, these form an equilateral triangle, yielding a hexagonal lattice. F) The sum of three plane waves at $60°$ interferes to form a hexagonal lattice if all waves are in phase (Eq. 5). G-I) Numerical simulation of pattern forming dynamics (Eq. 5). G) When $\tilde{\Sigma}$ is peaked near zero, pattern forming dynamics yield square grids. H) When $\tilde{\Sigma}$ is peaked far from zero, pattern forming dynamics yield quasi-periodic lattices comprised of many modes. I) A softened nonnegativity constraint induces a regular hexagonal lattice. Insets in G)-I) 2D Fourier transforms of the learned maps: 4 peaks, widely distributed activity, and 6 hexagonally distributed peaks, respectively.

This collection of results raises fundamental scientific questions. Why do these diverse architectures, across diverse tasks (both navigation and autoencoding), all converge to a grid-like solution, and what governs the lattice structure of this solution? We address this question by noting that all optimization problems in networks 1-3 contain within them a *common sub-problem*, which we call the position encoding objective: selecting hidden responses $G$ and linearly combining them with readout weights $W$ in order to predict place cell responses $P$ (Fig. 1A). We further show that due to the translation-invariance of place cell responses, the learning dynamics of this position encoding objective can be formulated as a pattern forming dynamical system, allowing us to understand the nature and structure of the resultant grid-like solutions and their dependence on various parameters.

## 3 Pattern formation theory predicts structure of learned representations

The common position encoding sub-problem identified in the previous section can be mathematically formulated as minimizing the following objective function

$$\mathcal{E}(G, W) = \|P - \hat{P}\|_F^2, \text{ where } \hat{P} = GW. \tag{1}$$

Here $P \in \mathbb{R}^{n_x \times n_P}$ represents true place cell activations, where $P_{x,i}$ is the activation of place cell $i$ at spatial index $x$. $G \in \mathbb{R}^{n_x \times n_G}$ represents hidden layer activations (which will learn grid-like representations), where $G_{x,j}$ is the activation of hidden unit $j$ at spatial index $x$. $W \in \mathbb{R}^{n_G \times n_P}$ represents linear readout weights, where $W_{ji}$ is the contribution of grid cell $j$ to place cell $i$. $\hat{P} \in \mathbb{R}^{n_x \times n_P}$ represents the predictions of the place cell encoding system. For simplicity, we consider an L2 penalty on encoding errors. Because we are ultimately interested in the hidden unit activations $G$, we replace $W$ with its optimum value for fixed $G$ (see App. B.1 for details):

$$\underset{W}{\operatorname{argmin}} \mathcal{E}(G, W) = (G^T G)^{-1} G^T P. \tag{2}$$

The objective $\mathcal{E}$ is unchanged by any transformation of the form $G \to GZ$, $W \to Z^{-1}W$. In particular, we can simplify our objective by choosing $Z$ so that $G$'s columns are orthonormal. Enforcing this constraint via Lagrange multipliers, we obtain the following Lagrangian

$$\mathcal{L} = \text{Tr}\left[G^T \Sigma G - \lambda(G^T G - I)\right], \tag{3}$$

where $\Sigma = PP^T$ is the $n_x \times n_x$ correlation matrix of place cell outputs. Note that assuming the place cell receptive fields uniformly and isotropically cover space, $\Sigma$ will, in the limit of large numbers of place cells, be approximately translation invariant (i.e. Toeplitz) and circularly symmetric. This Lagrangian is optimized when $G$'s columns span the top $n_G$ eigenvectors of $\Sigma$ (Eckart-Young-Mirsky, see App. B.2), and is invariant to a unitary transformation $G \to GU$. Moreover, since $\Sigma$ is a Toeplitz matrix, the eigenvectors of $\Sigma$ are approximately plane waves. Thus the optimization in (3) yields arbitrary linear combinations of different plane wave eigenmodes of $\Sigma$ corresponding to the $n_G$ largest eigenvalues. However, this multiplicity of solutions is a special feature due to the lack of any further constraints, like non-negativity. As we'll see below, once a nonlinear constraint, like non-negativity, is added, this multiplicity of solutions disappears, and the optimization favors a single type of map corresponding to hexagonal grid cells.

## 3.1 Single-cell dynamics

To build intuition, we begin by studying the case of a *single* encoding cell $g \in \mathbb{R}^{n_x}$ and difference of Gaussian place cell tuning. The Lagrangian for this cell is given by

$$\mathcal{L} = g^T \Sigma g + \lambda(1 - g^T g). \tag{4}$$

Gradient ascent on this objective function at fixed $\lambda$ yields the dynamics

$$\frac{d}{dt}g = -\lambda g + \Sigma g. \tag{5}$$

This is a **pattern forming dynamics** in which the firing fields at two positions $g_x$ and $g_{x'}$ mutually excite (inhibit) each other if the spatial autocorrelation $\Sigma_{xx'}$ of the desired place cell code at the two positions $x$ and $x'$ is positive (negative). Under this dynamics, patterns corresponding to the eigenmodes of largest eigenvalue of $\Sigma$ grow the fastest, with an exponential growth-rate given by the corresponding eigenvalue. In actuality, to solve the constrained optimization problem in (4) we run a projected gradient ascent algorithm in which we iteratively project $g$ to the constraint surface $g^T g = 1$. Such a dynamics converges to a linear combination of degenerate eigenmodes of $\Sigma$ all of whom share the same maximal eigenvalue. Since $\Sigma$ is translation invariant and circularly symmetric, these corresponding eigenmodes are linear combinations of plane waves whose wave-vectors lie on a ring in Fourier space whose radius $k^*$ is determined by the top eigenvalue-eigenvector pair of $\Sigma$.

In Fig. 2 we plot the eigenvalue associated to each plane wave as a function of its wave-vector for a Gaussian (A) and difference of Gaussian (B) place cell tuning curve. For Gaussian tuning, optimal wave-vectors, corresponding to the largest eigenvalues of $\Sigma$, lie close to the origin, while for difference of Gaussians tuning with covariance $\Sigma^*$, the optimal wave-vectors, corresponding to the largest eigenvalues of $\Sigma^*$, are concentrated on a ring of radius $k^*$ in Fourier-space, far from the origin.

Consistent with this analysis and the results of Fig. 1, numerical simulations of pattern forming dynamics corresponding to optimizing (4) yield quasi-periodic patterns like those in (Fig. 2G,H). In simulations, the finite box size discretizes Fourier space onto a lattice. Thus, numerical solutions will consist of discrete combinations of plane waves with wave-vectors of radius $k^*$. The lowest nonzero Fourier modes occur at $0°$ and $90°$ on the Fourier lattice. Therefore, when $\Sigma$'s spectrum is peaked near the origin as in the case of Gaussian place cell tuning (Fig. 2C), solutions will be dominated by square grids like those found in [10, 13]. Rings further from the origin may occasionally intersect six lattice points, "accidentally" yielding hexagonal grids like those observed in [14]. However, as the difference of Gaussian case shows, in general, optimal patterns can contain *any* mixture of wavevectors from a ring (Fig 2D), giving rise to amorphous patterns (Fig 2H; inset shows Fourier power is distributed over whole ring). Indeed, the encoding objective considered above does not prefer one type of lattice to another. As we will see, adding a nonnegativity constraint to our objective breaks this degeneracy, and reliably picks out hexagonal solutions.

## 3.2 A nonnegativity constraint favors hexagonal grids

We have seen empirically that a nonnegativity constraint tends to produce hexagonal grids (Fig. 1E). To understand this effect, we add a softened nonnegativity constraint to our objective function as follows

$$\mathcal{L} = g^T \Sigma g + \lambda(1 - g^T g) + \sigma(g), \tag{6}$$

where $\sigma(g)$ penalizes negative activites in the map $g$. It will be convenient to write $g_x$ as $g(\vec{x})$, treating $g$ as a scalar field defined for all points in space. Our objective then takes the form

$$\mathcal{L}[g(\vec{x})] = \iint_{\vec{x},\vec{x}'} g(\vec{x})\Sigma(\vec{x} - \vec{x}')g(\vec{x}') + \lambda\left(1 - \int_{\vec{x}} g^2(\vec{x})\right) + \int_{\vec{x}} \sigma(g(\vec{x})). \tag{7}$$

We can approximate the negativity penalty by Taylor expanding about 0: $\sigma(g) \approx \sigma_0 + \sigma_1 g + \sigma_2 g^2 + \sigma_3 g^3$. Our Lagrangian then has a straightforward form in Fourier space

$$\tilde{\mathcal{L}}[\tilde{g}(\vec{k})] \approx \int_{\vec{k}} |\tilde{g}(\vec{k})|^2 \tilde{\Sigma}(\vec{k}) + \tilde{\lambda}\left(1 - \int_{\vec{k}} |\tilde{g}(\vec{k})|^2\right)$$
$$+ [\sigma_0 + \sigma_1 \tilde{g}(\vec{0}) + \sigma_2 \int_{\vec{k}} |\tilde{g}(\vec{k})|^2 + \sigma_3 \iiint_{\vec{k},\vec{k}',\vec{k}''} \tilde{g}(\vec{k})\tilde{g}(\vec{k}')\tilde{g}(\vec{k}'')\delta(\vec{k} + \vec{k}' + \vec{k}'')]. \tag{8}$$

$\sigma_0, \sigma_1,$ and $\sigma_2$ will not qualitatively change the structure of the solutions. $\sigma_0$ simply shifts the optimal value of $\mathcal{L}$, but not its argmax; $\sigma_1$ controls the amount of the constant mode in the maps, and does not affect their qualitative shape; and $\sigma_2$ can be absorbed into $\tilde{\lambda}$ [17]. Critically, however, the cubic term $\sigma_3$ introduces an interaction between wavevector triplets $\vec{k}, \vec{k}', \vec{k}''$ whenever the three sum to zero (Fig. 2E).

In the limit of weak $\sigma_3$, the maps will be affected in two separate ways. First, weak $\sigma_3$ will pull the maps slightly outside of the linear span of the optimal plane-waves, or eigenmodes of $\Sigma$ of largest eigenvalue. As $\sigma_3 \to 0$, this effect shrinks and effectively disappears, so that we can assume the optimal maps are still constrained to be linear combinations of plane waves, with wave-vectors on the same ring in Fourier space. The second, stronger effect is due to the fact that no matter how small $\sigma_3$ is made, it will break $\mathcal{L}$'s symmetry, effectively forcing it to choose one solution from the set of previously degenerate optima. Therefore, in the limit of small $\sigma_3$, we can determine the optimal maps by considering which wavevector mixture on the ring of radius $k^*$ maximizes the nonlinear term

$$\mathcal{L}_{\text{int}} = \iiint_{\vec{k},\vec{k}',\vec{k}''} \tilde{g}(\vec{k})\tilde{g}(\vec{k}')\tilde{g}(\vec{k}'')\delta(\vec{k} + \vec{k}' + \vec{k}''). \tag{9}$$

Subject to the normalization constraint $\int |\tilde{g}(\vec{k})|^2 = 1$, this term is maximized when $\tilde{g}(\vec{k}) = \frac{1}{\sqrt{6}}\sum_{i=1}^{3}\delta(\vec{k} - \vec{k}_i) + \text{c.c.}[2]$, where $\vec{k}_1 + \vec{k}_2 + \vec{k}_3 = 0$. The only combination of $\vec{k}_1, \vec{k}_2, \vec{k}_3$ on the ring of radius $|k^*|$ that sums to zero is an equilateral triangle (Fig. 2E). Therefore, rather than arbitrary linear combinations of plane waves as in Eq. 1, the optimal solutions consist of three plane waves with equal amplitude and wavevectors that lie on an equilateral triangle.

$$g(\vec{x}) = \frac{1}{\sqrt{6}}(e^{i\vec{k}_1 \cdot \vec{x} + \phi_1} + e^{i\vec{k}_2 \cdot \vec{x} + \phi_2} + e^{i\vec{k}_3 \cdot \vec{x} + \phi_3} + \text{c.c.}). \tag{10}$$

The interaction $\mathcal{L}_{\text{int}}$ is maxizimed when $\phi_1 + \phi_2 + \phi_3 = 0$, in which case the three plane waves interfere to form a regular hexagonal lattice (Fig. 2F).

We can optimize the above Lagrangian using the same pattern forming dynamics as in Eq. 5, under the nonnegativity constraint $\sigma$ defined above[3]. When we perform numerical simulations of this dynamics, we find regular hexagonal grids (Fig. 2I). Taking the 2D Fourier transform of the resulting pattern reveals that the nonnegativity constraint has picked out three wavevectors oriented at $60°$ relative to one another (and their negatives) from the optimal ring of solutions (Fig. 2I, inset).

### 3.3 Hexagonal grids and $g \to -g$ symmetry breaking

We see from the above argument that the rectification nonlinearity is but one of a large class of nonlinearities which will favor hexagonal grids. A generic nonlinearity with a non-trivial cubic term in its Taylor expansion will break the $g \to -g$ symmetry, and introduce a three-body interaction which picks out hexagonal lattices. While nonnegativity is a specific nonlinearity motivated by biological considerations, a broad class of nonlinearities will achieve the same effect (numerical simulations in Appendix A1).

## 4 Multiple cells & nonnegativity yields hierarchies of hexagonal grids

We now return to the full Lagrangian with multiple grid cells (Eq. 3),

$$\mathcal{L} = \operatorname{Tr} G^T \Sigma G + \sum_{ij} \lambda_{ij} (I - G^T G)_{ij} + \sigma(G). \tag{11}$$

Recall that the solution space of the unperturbed objective with $\sigma = 0$ corresponds to maps $G$ whose wavevectors fall on a series of concentric rings in Fourier space, $\tilde{\Sigma}_{\text{top}}$ (Fig. 3B). By symmetry of $\mathcal{L}$, any unitary mixture of such a set of maps, $G \to GU$, will perform equally well. The nonlinearity $\sigma$ then breaks $U$-symmetry and selects for specific mixtures of wavevectors. As before, the cubic term $\sigma_3$ induces a three-body interaction which promotes triplets of wavevectors that sum to zero.

If the number of maps to be learned $n_G$ is small enough that the top subspace rings $\tilde{\Sigma}_{\text{top}}$ form a relatively thin annulus in Fourier space, then all wavevector triplets that sum to 0 will be approximately equilateral, giving rise to regular hexagonal grid maps. Once the number of maps to be learned is large, the top subspace rings will form a thick annulus, inside of which many different triplet wavevector arrangements - not just equilateral triangles - will sum to 0. Despite this, a significant fraction of maps learned in simulations are still hexagonal. As a first step toward understanding these results, we analyze a few possible non-equilateral triplets and show that the optimum has a dominant equilateral component.

One simple non-equilateral arrangement is any triplet of the form $(\vec{k}, \vec{k}, -2\vec{k})$, corresponding to a stripe pattern with its first overtone. In Appendix B.3, we prove that such an arrangement contributes at most $\mathcal{L}_{\text{int}}^{\text{coupled}} = 3/2$ to the Lagrangian, whereas an equilateral triangle contributes $\mathcal{L}_{\text{int}}^{\text{decoupled}} = 2 \times 2 \times 3!/6^{3/2} \approx 1.63$. Another possibility is a *hybrid* map consisting of a mixture of two equilateral triangles, one twice as big as the other. We prove that the optimal mixture puts most weight on either the big *or* the small triangle, making the optimal solutions relatively pure-tone hexagonal maps.

Empirically, we can optimize the unperturbed multiple grid cell Lagrangian in Eq. 3 using the pattern forming dynamics

$$\frac{d}{dt} G = -\lambda G + \Sigma G \tag{12}$$

and enforcing orthonormality of $G$. Introducing the nonnegativity constraint, we obtain dominantly hexagonal maps across multiple spatial scales (Fig. 3D).

Historically, the roughly constant ratio of grid scale in neighboring MEC modules has led to interest in geometric hierarchies - both which kinds of encoding objectives favor them, and which pattern forming dynamics produce them. During simulations, we find that lattice discretization effects can sometimes create the illusion of a geometric hierarchy (3E). Roughly speaking, if $\tilde{\Sigma}$ is peaked at the origin, as the number of encoding maps is increased, wavevector rings are filled up one by one. Due

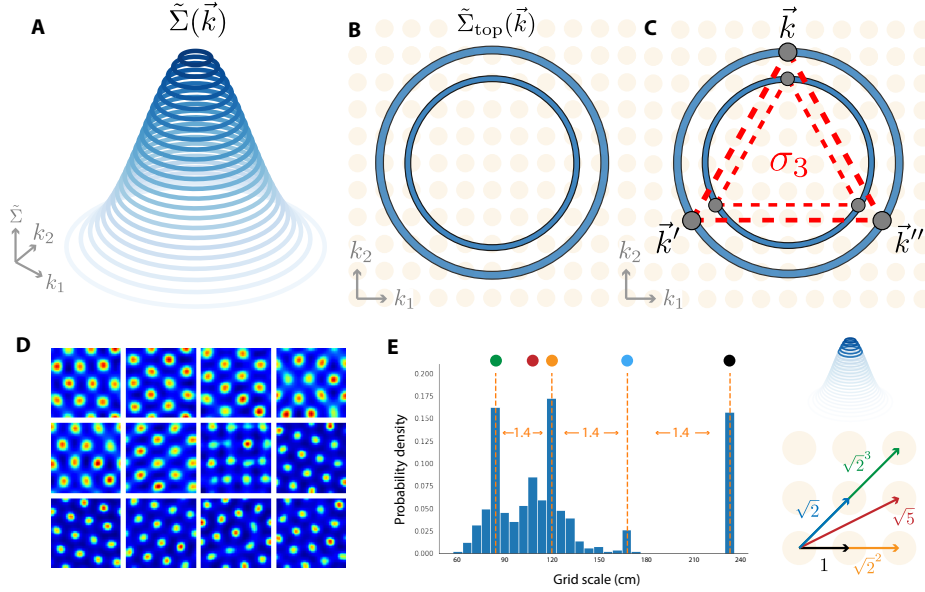

Figure 3: A) Visualization of $\tilde{\Sigma}$. B) Top subspace of $\Sigma$ when multiple grid cells are available. Absent any other constraint, there is a full rotational degree of freedom within this space. C) A cubic term in the nonlinearity induces a three-body interaction between triplets of spatial frequencies which add to zero. D) Results of multi-grid pattern forming dynamics with nonnegativity constraint show regular hexagonal grids across multiple spatial scales. E) Left: distribution of grid scales for pattern forming dynamics with Gaussian place cell tuning curve. This distribution can create the *illusion* of geometric hierarchy, but is due to discretization restricting the lowest frequency modes to the lattice. Right: (top) First five available spatial frequencies and (bottom): their corresponding wavevectors. Three of the first four lattice spacings are separated by a ratio of $\sqrt{2} \approx 1.4$.

to the geometry of the lattice points in wave-vector space, the five wave-vectors of smallest length (i.e. smallest spatial frequency) will have relative length ratios given by $(1, \sqrt{2}, \sqrt{2}^2, \sqrt{5}, \sqrt{2}^3)$. These correspond exactly to the peaks we observe in 3E. Because these particular scale relationships are strongly dependent on discretization effects and boundary conditions - both of which arise from an ad hoc modelling decision to use a square, periodic environment - it is not clear that this effect in the model is a potentially likely explanation for the apparent geometric hierarchy of maps in MEC.

## 5 Unifying normative and mechanistic models of grid cells

This theory establishes a remarkable *equivalence* between optimizing the position encoding objective, and the recurrent neural network dynamics of continuous attractor models. To see this, consider a closely related single-cell Lagrangian $\bar{\mathcal{L}}$ obtained by adding a constraint on the sum of the cell activations:

$$\bar{\mathcal{L}} = g^T \Sigma g - \lambda g^T g + \mu 1^T g \tag{13}$$

Empirically, we find that that this sum constraint minimally affects the structure of the optimal patterns. We can optimize this objective for nonnegative $g$ by stepping along $\bar{\mathcal{L}}$'s gradient and rectifying the result. In the limit of small step size, this becomes

$$\frac{d}{dt}g = \begin{cases} -\lambda g + \Sigma g + \mu & g > 0 \\ -\lambda g + \sigma[\Sigma g + \mu] & g = 0 \end{cases} \tag{14}$$

where $\sigma$ is the rectifying nonlinearity. Interestingly, this is *almost exactly* the dynamics proposed in continuous attractor models of grid cells [4, 5, 7, 6], with appropriate choice of time constant $\tau$ and scaling of the recurrent weights $J$ and feedforward drive $b$:

$$\tau \frac{d}{dt} g = -g + \sigma[Jg + b]. \tag{15}$$

We prove in App. B.4 and B.5 that the dynamics of Eq. 14 and Eq. 15 have identical fixed points which satisfy KKT optimality conditions for the constrained position encoding objective. In this unification of normative and mechanistic models, the spatial autocorrelation structure $\Sigma$ of the desired place cell code in a normative position encoding model corresponds to the recurrent connectivity $J$ of the mechanistic RNN model. While the normative model learns grid cell firing fields as a function of space by minimizing an encoding objective, the mechanistic RNN model generates stable periodic bump patterns on a sheet of neurons by choosing translation invariant connectivity between neurons on the sheet. Our theory predicts that if the neural nonlinearity ReLU in a mechanistic model breaks firing rate inversion symmetry $g \rightarrow -g$, then periodic patterns on the neural sheet should be hexagonal. Historically, a rectifying nonlinearity has indeed been used in mechanistic models, and hexagonal grids have emerged. Consistent with our theory, other nonlinearities that preserve the firing rate inversion symmetry yield square grids (see Appendix A1).

This unifying connection between normative and mechanistic models yields an intuitive insight: continuous attractor dynamics not only reproduce the patterns of activity observed in the MEC; they are *equivalent* to optimizing the position encoding objective. That is, the patterns formed in the continuous attractor model are also optimal for linearly generating place cell-like activations.

## 6 Discussion

Our unifying theory and numerical results suggest that much of the diversity and structure of solutions obtained across many different normative models of MEC can be explained by the learning dynamics of a simple linear, place cell encoding objective. This is intriguing given that the architectures and navigational tasks employed in [13, 14] are considerably more sophisticated. Further studies could explore how changing this common subproblem changes the solutions found by RNNs trained to navigate. Moreover, our theory predicts *why* hexagonal grids should emerge in RNNs trained to path integrate, but it does not explain *how* RNNs trained via backprop learn to stabilize and update these patterns in order to path integrate. Future studies could reverse engineer these trained networks to determine whether their inner workings coincide with the simple mechanistic models we describe in Section 5 and have been proposed for the past 20 years.

Furthermore, our theory made simplifying assumptions about uniform, isotropic coverage of place cell representations, yielding highly regular, stable grid cell patterns by solving the position encoding objective. Our theoretical framework enables us to explore quantitatively how grid cell solutions change when the environment is deformed, rewards or obstacles are incorporated, or place cells are lesioned. Recent experiments have characterized the MECs response to each of these scenarios [18, 19, 20, 21], but no unified theory has been put forth to explain the results. These questions could potentially be addressed by drawing from a rich literature of how patterns respond to defects in either spatial statistics or neural connectivity. Such defects could play a role in accounting for the heterogeneity of grid cells [22]. Another interesting approach is to incorporate landmark inputs in trained networks, in addition to velocity inputs. Such landmark inputs are known to correct drift in the grid cell system [23] and can successfully account for various deformations in grid cell firing patterns due to environmental manipulations [8, 9].

Finally, a growing body of work has explored experimentally the hypothesis that MEC encodes continuous variables other than position, such as sound pitch [24] or abstract quantities like the width and height of a bird [25]. While we have referred to a "position" encoding objective and "path" integration, we note that our theory actually holds for *generic* continuous variables. That is, we would expect networks trained to keep track of sound pitch and volume to behave the same way. Perhaps, intriguingly, grid like structure may be relevant for neural processing in even more abstract domains of semantic cognition [25, 26]. Overall, the unifying pattern formation framework we have identified, that spans both normative and mechanistic models, affords a powerful conceptual tool to address many questions about the origins, structure, variability and robustness of grid-like representations in the brain.

## Acknowledgments

S.G. thanks the Simons, and James S. McDonnell foundations, and NSF Career 1845166 for funding. B.S. thanks the Stanford Graduate Fellowship for financial support.

## Footnotes

[1]Though several works have shown that if grid cells do obey a lattice-like structure, hexagonal lattices are better than other lattice types for decoding position under noise [15] and are more economical [16].

[2]Note that c.c. is shorthand for complex conjugate. For any real solution $g(\vec{x})$ to Eq. 6, $\tilde{g}(\vec{k}) = \tilde{g}^{\dagger}(-\vec{k})$. Therefore, for each wavevector $\vec{k}$ we must also include its negative, $-\vec{k}$.

[3]In App. B.4 we prove that the dynamics satisfies KKT optimality conditions for the Lagrangian in Eq. 6.

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
