[Supplementary Material]

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

# A    Pattern formation

Figure 4: Stable pattern formed in the neural sheet under continuous attractor dynamics (Eq. 15) for a nonlinearity $\sigma'(x)$ which breaks the firing rate inversion symmetry, and a nonlinearity which preserves the firing rate inversion symmetry. In the dynamics, the nonlinearity $\sigma'(x)$ is given by the derivative of the nonlinearity in the Lagrangian. Since the derivative of an even function is odd, we expect squares (or quasi-periodic patterns) when $\sigma'(x)$ is odd, and hexagons when it is not. Simulations confirm that for a non-odd nonlinearity $\sigma'(x) = \text{relu}(x)$ (left), the pattern converges to hexagons. For an odd nonlinearity $\sigma'(x) = \tanh(x)$ (right), the pattern converges to squares.

$$\sigma(x) = x^3 \qquad \sigma(x) = e^x$$

Figure 5: Nonlinearities which break the firing rate inversion symmetry $g \to -g$ favor hexagons. Shown above are numerical simulations of pattern forming dynamics (Eq. 6) with the choices $\sigma(x) = x^2$ (left) and $\sigma(x) = e^x$ (right).

Figure 6: Training with an extremely sharply peaked place cell tuning curve (1cm$^2$) yields amorphous, quasi-periodic patterns like those in [14]. This result can be understood within our theory by noting that the place cell tuning curve is approximately a delta function in real space: (A) shows an example of the place cell activity vector at a given location in space. Only one place cell has nontrivial activiation. (B) The place cell covariance $\Sigma$ is thus extremely sharply peaked, and yields an extremely *broad* profile in Fourier space, $\tilde{\Sigma}$. Therefore, all low frequency modes are approximately *equivalent* under the decoding objective, and solutions consist of linear combinations of all low frequency modes, yielding quasi-periodic patterns like those in (C,D).

# B    Optimal position encoding

**Lemma B.1.** *The optimal $W$ of the objective*

$$\mathcal{E}(G, W) = \|P - \hat{P}\|_F^2, \text{ where } \hat{P} = GW. \tag{16}$$

*for fixed $G$ is given by $W = (G^T G)^{-1} G^T P$.*

*Proof.* Extemizing the objective $\mathcal{E}$ with respect to $W$ for fixed $G$ gives

$$\frac{\partial \mathcal{E}(G, W)}{\partial W} = \frac{\partial}{\partial W} \|P - GW\|_F^2 = 2G^T(P - GW) = 0 \tag{17}$$

$$\implies W = (G^T G)^{-1} G^T P \tag{18}$$

$\square$

**Theorem B.2.** *(Eckart-Young-Mirsky) The objective $Tr(G^T \Sigma G)$ for $G^T G = I$ is maximized when $G$ spans the top eigenvectors of $\Sigma$.*

*Proof.*

$$\mathrm{Tr}(G^T \Sigma G) = \mathrm{Tr}(\Sigma G G^T) \leq \sum_{i=1}^{T} \lambda_i(\Sigma) \lambda_i(G G^T) \tag{19}$$

Where we have used the Von-Neumann trace inequality, and $\lambda_i(A)$ gives the eigenvalues of $A$ in descending order. Notice that $G G^T$ is a projection operator of rank $n_G$, since $(G G^T) G = G$. Therefore, $\lambda_i(G G^T) = 1$ for $i = 1, \ldots, n_G$, and $\lambda_i(G G^T) = 0$ for $i = n_G + 1, \ldots, T$. Thus we have

$$\mathrm{Tr}(G^T \Sigma G) \leq \sum_{i=1}^{n_G} \lambda_i(\Sigma) \tag{20}$$

This bound is saturated when $G$ spans the top eigenvectors of $\Sigma$. $\square$

**Lemma B.3.** *The multiple grid cell Lagrangian Eq. 11 with nonlinearity $\sigma$ favors a hierarchy of hexagonal grids.*

Figure 7: A) Visualization of labels of wavevectors (a1,a2,a3,b1,b2,b3) on two concentric rings in Fourier space B) The interaction given by a pair of cells spanning $\vec{k}_{a1}, \vec{k}_{b1}$ (Eq. 21). C) The interactions given by a pair of cells each constrained to one ring (Eq. 23). D) The interactions given by the hybrid solution (Eq. 24).

*Proof.* The cubic term $\sigma_3$ in the nonlinearity will induce a three-body interaction which picks out triplets of wavevectors that sum to zero. The simplest triplet of wavevectors that sum to zero is an equilateral triangle on a single ring, corresponding to hexagonal grids (Fig. 3C). However, the multiple grid cell Lagrangian permits other combinations. Consider the case where one eigenvector of $\Sigma$ has wavevector $\vec{k}_{a1}$, and another eigenvector has wavevector $\vec{k}_{b1} = 2\vec{k}_{a1}$. Then the nonlinearity

introduces a cubic contribution that couples multiple rings, using the combination $\vec{k}_{b1} - \vec{k}_{a1} - \vec{k}_{a1} = 0$. This combination would yield a pattern consisting of a stripe and its harmonic. However, this pattern does not appear in simulation, and indeed is sub-optimal in the position encoding Lagrangian. To see this, we can parameterize a pair of cells by

$$\tilde{g}_A(\vec{k}) = \frac{1}{\sqrt{2}}\left(\cos\theta\delta(\vec{k} - \vec{k}_{a1}) + \sin\theta\delta(\vec{k} - \vec{k}_{b1}) + \text{c.c.}\right),$$
$$\tilde{g}_B(\vec{k}) = \frac{1}{\sqrt{2}}\left(-\sin\theta\delta(\vec{k} - \vec{k}_{a1}) + \cos\theta\delta(\vec{k} - \vec{k}_{b1}) + \text{c.c.}\right), \tag{21}$$

Then the contribution to the cubic term in the objective from this orthogonal pair is given by

$$\mathcal{L}_{\text{int}} = \frac{6}{\sqrt{2}^3} \times \left(\cos\theta^2\sin\theta + \cos\theta\sin\theta^2\right) \tag{22}$$

where the 6 arises from combinatorics. The maximum of this contribution is given by $\mathcal{L}_{\text{int}}^{\text{coupled}} = 3/2$ at $\theta = \pi/4$. On the other hand, the maximum contribution from a pair of cells that each stay within a single ring:

$$\tilde{g}_A(\vec{k}) = \frac{\delta(\vec{k} - \vec{k}_{a1}) + \delta(\vec{k} - \vec{k}_{a2}) + \delta(\vec{k} - \vec{k}_{a3})}{\sqrt{6}} + \text{c.c.},$$
$$\tilde{g}_B(\vec{k}) = \frac{\delta(\vec{k} - \vec{k}_{b1}) + \delta(\vec{k} - \vec{k}_{b2}) + \delta(\vec{k} - \vec{k}_{b3})}{\sqrt{6}} + \text{c.c.} \tag{23}$$

where $\vec{k}_{a1}, \vec{k}_{a2}, \vec{k}_{a3}$ form an equilateral triangle within the inner ring and $\vec{k}_{b1}, \vec{k}_{b2}, \vec{k}_{b3}$ form an equilateral triangle within the outer ring is given by $\mathcal{L}_{\text{int}}^{\text{uncoupled}} = 2 \times 2 \times 3!/6^{3/2}$. Noting that $\mathcal{L}_{\text{int}}^{\text{coupled}} = 1.5 < \mathcal{L}_{\text{int}}^{\text{uncoupled}} \approx 1.63$, we see that the solution which uncouples rings is optimal for the encoding Lagrangian compared to the solution involving overtones of stripes.

Next, we consider a hybrid solution consisting of a superposition of *triplets* over multiple rings, i.e.,

$$\tilde{g}_A(\vec{k}) = -\sin\theta\frac{\delta(\vec{k} - \vec{k}_{a1}) + \delta(\vec{k} - \vec{k}_{a2}) + \delta(\vec{k} - \vec{k}_{a3})}{\sqrt{6}} + \cos\theta\frac{\delta(\vec{k} - \vec{k}_{b1}) + \delta(\vec{k} - \vec{k}_{b2}) + \delta(\vec{k} - \vec{k}_{b3})}{\sqrt{6}} + \text{c.c.},$$
$$\tilde{g}_B(\vec{k}) = \cos\theta\frac{\delta(\vec{k} - \vec{k}_{a1}) + \delta(\vec{k} - \vec{k}_{a2}) + \delta(\vec{k} - \vec{k}_{a3})}{\sqrt{6}} + \sin\theta\frac{\delta(\vec{k} - \vec{k}_{b1}) + \delta(\vec{k} - \vec{k}_{b2}) + \delta(\vec{k} - \vec{k}_{b3})}{\sqrt{6}} + \text{c.c.} \tag{24}$$

A simple calculation gives the contribution of this interaction to the encoding Lagrangian

$$\mathcal{L}_{\text{int}}^{\text{hybrid}} = \frac{2 \times 3 \times 3}{\sqrt{6}^3}\left(\cos^2(\theta)\sin(\theta) + \sin^2(\theta)\cos(\theta)\right) + \frac{2 \times 2 \times 3!}{\sqrt{6}^3}\cos^3(\theta) \tag{25}$$

which has an optimum of 1.86 at $\theta \approx .108 \times \pi$, yielding only a small mixing between modes of different length scales. A plot of this overtoned solution is shown in Fig. 8, and is nearly indistinguishable from the decoupled solution. It may be interesting to investigate whether such overtoned patterns exist in empirical data in the MEC.

□

**Lemma B.4.** *The projected gradient ascent dynamics in (Eq. 14) satisfies KKT optimality conditions for the position encoding Lagrangian with a sum constraint.*

*Proof.* The sum constrained problem can be written as

Figure 8: A) Pure hexagonal lattice with no overtones. B) Overtoned hexagonal lattice when one grid cell spacing has twice the periodicity of another (Eq. 24).

$$\max_{g} g^T \Sigma g \tag{26}$$

$$s.t. \quad \begin{matrix} g^T g = a \\ g^T \vec{1} = b \\ g \geq 0 \end{matrix} \tag{27}$$

The KKT optimality conditions for this problem are

$$\begin{aligned} \Sigma g - \lambda g + \mu \vec{1} + \eta &= 0 \\ g^T g &= a \\ g^T \vec{1} &= b \\ g &\geq 0 \\ \eta &\geq 0 \\ \eta \circ g &= 0 \end{aligned} \tag{28}$$

Fixed points of the dynamics in Eq. 14 satisfy

$$0 = \begin{cases} -\lambda g + \Sigma g + \mu & g > 0 \\ -\lambda g + \text{ReLU}\big[\Sigma g + \mu\big] & g = 0 \end{cases} \tag{29}$$

which is equivalent to

$$\begin{matrix} -\lambda g_i + (\Sigma g)_i + \mu = 0 & \text{if} & g_i > 0 \\ -\lambda g_i + (\Sigma g)_i + \mu \leq 0 & \text{if} & g_i = 0 \end{matrix} \tag{30}$$

This can be written as

$$-\lambda g_i + (\Sigma g)_i + \mu + \eta_i = 0 \tag{31}$$

where $\eta_i \geq 0$ for all $i$ and $\eta_i = 0$ whenever $g_i > 0$. Thus all of the following conditions are satisfied:

$$-\lambda g + \Sigma g + \eta + \mu = 0 \tag{32}$$
$$g \geq 0 \tag{33}$$
$$\eta \geq 0 \tag{34}$$
$$\eta \circ g = 0 \tag{35}$$

Appropriate choice of $\lambda$ and $\eta$ will ensure $g^T g = a$ and $g^T \vec{1} = b$, completing the KKT conditions. Thus, the fixed points of these dynamics are optima of the encoding problem.

$\square$

**Lemma B.5.** *The fixed points of the mechanistic attractor dynamics in Eq. 15 and the projected gradient ascent dynamics of Eq. 14 are identical for appropriate choice of $\tau, J, b$.*

*Proof.* Set $\tau = 1, J = \Sigma/\lambda, b = \mu/\lambda$. The fixed point conditions of Eq. 15 and Eq. 14 are, respectively

$$0 = -\lambda g + \text{ReLU}[\Sigma g + \mu]. \tag{36}$$

$$0 = \begin{cases} -\lambda g + \Sigma g + \mu & g > 0 \\ -\lambda g + \text{ReLU}[\Sigma g + \mu] & g = 0 \end{cases} \tag{37}$$

For components where $g = 0$, these are identical. For components where $g > 0$, the two conditions become

$$\begin{aligned} \lambda g &= \text{ReLU}[\Sigma g + \mu] \\ \lambda g &= \Sigma g + \mu \end{aligned} \tag{38}$$

which are equivalent since $g > 0$ implies $\text{ReLU}[\Sigma g + \mu] = \Sigma g + \mu$.

$\square$

## C   Training details for normative models

### C.1   Simulated trajectory statistics

Place cell receptive field centers $\vec{c}_i$, $i = 1, \dots, n_P$ are distributed randomly and uniformly over a $(2.2m \times 2.2m)$ environment. The response of the $i^{th}$ place cell is simulated using either a Gaussian tuning curve, $p_i(x) = e^{-||x-c_i||^2/2\sigma^2}$, or a difference of Gaussians tuning curve, $p_i(x) = e^{-||x-c_i||^2/2\sigma_1^2} - e^{-||x-c_i||^2/2\sigma_2^2}$ where $x$ is the current location of the animal. Animal trajectories are generated using the rat motion model described in [27]. We collect the place cell activations at $n_x$ locations as the animal explores its environment in a matrix $P \in \mathbb{R}^{n_x \times n_P}$.

### C.2   1-layer neural network

We train a 1-layer neural network with $n_G$ hidden units via an unsupervised Hebbian learning algorithm on place cell inputs $P$. The synaptic weights $W \in \mathbb{R}^{n_G \times n_P}$ are updated according to the following generalized Hebbian algorithm:

$$W_{ij}^{t+1} = W_{ij}^t + \varepsilon \left( g_i^t p_j^t - g_i^t \sum_{k=1}^{i} W_{kj}^t g_k^t \right) \tag{39}$$

The first term in parentheses can be seen as the simple Hebbian learning rule, while the second term implements a Gram-Schmidt like orthogonalization. As noted in [10], this training procedure is equivalent to performing PCA on the matrix $P$. We perform PCA on $P$ with $n_G = 9$ components to obtain the results in Fig. 1C,D (left).

To implement the nonnegativity constraint, we simply perform non-negative matrix factorization (NNMF) on the matrix $P$ with $n_G = 9$ components, using a standard NNMF toolbox. The result is shown in Fig. 1E (left).

### C.3   RNN path integrator network

The task and training protocol are replicated from [14]. Velocity signals from the simulated trajectory are given as input, and the network is trained to produce estimates of the simulated place cell activities as output. The network has a vanilla RNN architecture: 2 linear input units for x- and

y-velocity, a set of recurrently connected hidden units, and linear readout units. Network update and place cell estimate equations are as follows:

$$r^{t+1} = \sigma \left[ Jr^t + M\vec{v}^t \right] \tag{40}$$

$$p^t = Wr^t \tag{41}$$

where $r$ is the vector of neuron activities, $J$ the matrix of recurrent weights, $M$ is the network's velocity input weights, $\vec{v}$ is the agent's 2-dimensional velocity in the arena, $\sigma$ is a pointwise tanh nonlinearity, $p$ is the vector of estimated place cell activities, and $W$ is the place cell readout weights.

During training, trajectories of steps were generated in batches of 200 and fed to the path integrator network. We used the RMSProp optimizer to minimize the cross-entropy between the estimated and true place cell activities.

A summary of the task, architecture, and training parameters is given below:

| Task | |
|---|---|
| Arena size | (2.2m x 2.2m) |
| Average agent speed | 0.1 m/sec |
| # place cells | 512 |
| place cell $\sigma_1, \sigma_2$ | 20cm, 40cm |
| **Architecture** | |
| # RNN units | 512 |
| Input | $(v_x, v_y)$ |
| **Training** | |
| Path length | 50 |
| Batch size | 200 |
| Number of batches | 10000 |
| Optimizer | RMSProp |
| Learning rate | 1e-4 |

To implement the non-negativity constraint, we swap out the RNN nonlinearity from a tanh to a ReLU.

## C.4   LSTM path integrator network

The task and training protocol were identical to that of the RNN described above. The model architecture was reproduced from [14], consisting of x- and y-velocity inputs to an LSTM with 128 units, followed by a linear layer of 512 units (which the authors called the "g-layer"), followed by a final readout to the estimated place cell activities. In order to match [14], we train with L2 regularization on decoder weights $W$, and Dropout at a rate of 0.5 on the g-layer units, which the authors claimed were necessary to achieve grid-like results. A summary of the task, architecture, and training parameters is given below:

| Task | |
|---|---|
| Arena size | (2.2m x 2.2m) |
| Average agent speed | 0.1 m/sec |
| # place cells | 512 |
| place cell $\sigma_1, \sigma_2$ | 20cm, 40cm |
| **Architecture** | |
| # LSTM units | 128 |
| # g-layer units | 512 |
| Input | $(v_x, v_y)$ |
| **Training** | |
| Path length | 50 |
| Batch size | 200 |
| Number of batches | 10000 |
| Optimizer | RMSProp |
| Learning rate | 1e-4 |
| Dropout rate | 0.5 |
| Decoder weight regularization | 1e-5 |

To implement the non-negativity constraint, we swap out the LSTM nonlinearity from a tanh to a ReLU, and include a ReLU nonlinearity in the dense layer from LSTM units to g-layer units.