[Reviews · NeurIPS 2019]

Reviewer 1



Originality: This is a highly original manuscript which goes beyond using known algorithms to study technically challenging problems and instead attempts to explain mechanisms of fundamental neuro-systems in the MEC and HCF. Quality: The claims, while profound, appear to be well built upon the 3 main figures in the article. The authors acknowledge that further study could be done along the lines of quantitatively how grid cell solutions change when the environment is deformed, rewards or obstacles are incorporated, or place cells are lesioned, although few limitations are actually noted and overall strengths are highlighted in the article. Clarity: The article is well written and easy to understand even supposing the readers have little knowledge of place/grid cells. Significance: The article attempts to advance science in a theoretical sense which could have impact on future work both theoretical and applied given the current neuromodulation attempts to restore/improve human memory (and subservient to those navigation) systems

Reviewer 2



-The paper is clearly written. The figures are well-thought-out and nicely complement the explanations provided in the text. I find the presentation of some of the material in the SI slightly odd. Some of them are standard results in linear algebra (or follow directly from standard results), but they read like they are "new results". I assume the intent is to make the paper as self-contained as possible but please make this clear. -The findings appear to be technically sound. The analytical results are supported by appropriate simulations. - While the normative framework utilized in the work is not new, the analysis that establishes the conditions under which hexagonal grid cells emerge is original and interesting and I believe relevant to NeurIPS. -In the introduction, it is claimed that there is no theoretical clarity on when and why grid cells emerge (l42-43). This is not quite true. [a] (and relatedly reference [17] ) has shown that hexagonal grid cells are optimal because they maximize spatial resolution. Please discuss. In your experiments, do you find that hexagonal lattices provide better reconstruction (e.g relative to square)? Relevant citations: [a] Mathis, Stemmler, Herz, eLife, 2015, Probable nature of higher-dimensional symmetries underlying mammalian grid-cell activity patterns, [b] Whittington et al, NeurIPS, 2018, another related normative explanation of grid cells as basis functions for encoding transition statistics. ___________ Post author's feedback update: Thanks for your response. Please include relevant discussion from your response in the final manuscript. I've increased my score.

Reviewer 3



The work addresses the issue of how grid-patterned spatial activity in medial entorhinal cortex might arise as a function of learning using hippocampal place cell inputs in simulated random-walk experiments. This is not the first work to do so, but does take such efforts significantly forward in comparing and contrasting multiple approaches and providing a key means by which such networks can be generated to produce hexagonal versus square grid patterns. The work is presented quite clearly and in detail. In terms of significance, it would seem that the entire field of researchers working on this problem fail to consider that grid-patterning in medial entorhinal cortex is not "learned" during random explorations of the environment, but develops in the absence of any significant such behavior. Furthermore, even if such patterning were "learned", it would almost certainly involved many more sources of information (e.g., head direction cell input and egocentric border cell input). While my comments call into question the value of the work in this field to neuroscience, it seems that the authors have done an admirable job of addressing the questions as they stand and the work may prove useful to non-neuroscientific applications.

[Author Response · NeurIPS 2019]

We thank all reviewers for careful reviews and many positive comments, including **R1**: "a highly original manuscript
which goes beyond using known algorithms to study technically challenging problems and instead attempts to explain
mechanisms of fundamental neuro-systems", "...providing an important critical analysis for gaps in the neuroscience
literature", "could have impact on future work both theoretical and applied"; **R3**: "original and interesting and I believe
relevant to NeurIPS", "...clearly written. The figures are well-thought-out" **R4**: "take[s] [previous] efforts significantly
forward", "the authors have done an admirable job". We now respond to the other reviewer feedback:

**Response to Reviewer 3:** We thank **R3** for pointing out a pair of previous works deriving grid cells from a normative
perspective. These are nice papers and we will add them to the many normative grid cell papers we already cited. We
note however significant differences in focus between these previous works and ours. For example Whittington et. al.
2018 derived grid cells by learning basis functions for local transition operators, but this approach yielded square, not
hexagonal grid cells. Mathis et. al. eLife 2015 demonstrated that the Fisher information (FI) of independent Poisson
firing hexagonal grid cells is larger than that of square grid cells by a factor of $1.15$ under certain assumptions about
firing field width. However, they did not address at all the question of how a neural network could convert velocity
signals into spatial signals, and under what conditions such a neural network would specifically use intermediate
representations resembling hexagonal grid cells. They simply *assumed* the existence of hexagonal firing fields and
computed their FI. It is this latter question about neural networks that is the primary focus of our work; we feel it
is important because previous results exploring how neural networks can convert velocity signals to spatial signals,
yielded conflicting results: i.e. our Ref. 10 yielded square grid cells, while our Ref. 11, published in Nature, yields
very noisy hexagonal cells, but as we demonstrate in this paper, yield very robust and clean *square* grid cells if *one*
parameter (batch size) is changed. Indeed this in and of itself is a new result in our paper. Our main contribution, which
has never been done in any prior work, involves connecting neural network training to the theory of pattern formation,
thereby providing a unifying framework to reconcile these conflicting results as well as yield, to our knowledge for the
first time, a robust method for generating *hexagonal* grid cells in a square box as intermediate representations in neural
networks that convert velocity signals to spatial signals. We do however, take **R3**'s excellent suggestions to heart, and
we will expand our introduction to discuss more carefully, and in a more toned down manner, the diverse and interesting
contributions of previous normative works and explain how our or work relates to and is different from this previous
work. We will also provide full training details and code to ensure re-producibility of our work.

**R3** also raises an interesting point that hexagonal grid cells have higher FI than square ones. We do not see appreciable
differences in positional coding between our square and hexagonal grid cell networks. There are two good theoretical
reasons to believe such differences, if any, should be small. First, based on considerations of Mathis et. al. eLife 2015
the FI of *both* hexagonal and square grid cell networks is $O(N)$ where $N$ is the number of grid cells. Thus, for large
$N$, the spatial resolution (inverse square root of FI) of both codes will be much more similar to each other relative to
the scale of either the box or the mouse itself (i.e compare $O(1/\sqrt{N})$ and $O(1/\sqrt{1.15N})$ resolution for square and
hexagonal grids respectively to the $O(1)$ spatial scale of the mouse. Second, the dominant source of inaccuracies in
the estimate of position in path integrator neural networks arises from imperfections or noise in the mechanism that
integrates velocity to construct position. For example, Hardcastle et. al. Neuron 2015 showed errors in decoded position
from both mouse and rat entorhinal cortex rose rapidly with time since a border encounter (which can correct position
estimates). Thus the FI based theory of Mathis et. al. eLife 2015 cannot address this dominant source of error, as it
simply *assumes* the existence of perfect hexagonal firing fields, without ever considering additional errors arising from
imperfections in the very neural network mechanisms required to construct such firing fields from velocity signals. So
based on these two good reasons, our preliminary analysis that square and hexagonal grid cells perform similarly is not
inconsistent with Mathis et. al. eLife 2015. However, before publication in NeurIPS, we will follow up much more
carefully on **R3**'s very interesting suggestion by doing extensive analyses to detect $O(1/\sqrt{N})$ differences in position
estimation between square and hexagonal grid networks and we will report what we will find.

**Response to Reviewer 4:** We very much appreciate **R4**'s general concern about the field that grid cells may not be
learned from environmental statistics but rather could be innate before spatial experience. In fact, our prejudice is
with the latter viewpoint, in agreement with the reviewer. We emphasize, that this paper, as we have written it, is
completely agnostic to this issue, as we do not view our backpropagation training as a model of the *learning* process in
the actual mouse. Rather, backpropagation to us (as in many other works applying task based neural network training to
neuroscience) is simply a method of exploring the space of neural network solutions to the problem of path integration
(converting velocity to spatial representations) to achieve theoretical clarity on when hexagonal grid cell representations
emerge. We find it exciting that they emerge very naturally, simply when firing rates are positive. We make no claims
of course, that the mouse learns them this way. And indeed, given the prevalence of grid cell solutions found by our
path integrator networks, our work suggests innate formation of grid cell representations before spatial experience may
actually facilitate path integration during spatial experience.

**Response to Reviewer 2:** We thank you for uniformly positive comments and we will provide code upon publication.

[Meta-Review · NeurIPS 2019]

This paper explains theoretically how hexagonal grid patterns can occur from simple constraints. Reviewers uniformly liked this paper and it is an easy accept. The work was viewed as solid but somewhat incremental contribution. It would have been selected for an oral if it discussed the connection to AI/ML and/or made testable predictions and verified them.